

# Developing a climatological simplification of aerosols to enter the cloud microphysics of a global climate model

Ulrike Proske[1], Sylvaine Ferrachat[1], and Ulrike Lohmann[1]

[1]Institute for Atmospheric and Climate Science, ETH Zürich, Zürich, Switzerland

**Correspondence:** Ulrike Proske (ulrike.proske@env.ethz.ch)

**Abstract.** Aerosol particles influence cloud formation and properties. Hence climate models that aim for a physical representation of the climate system include aerosol modules. In order to represent more and more processes and aerosol species, their representation has grown increasingly detailed. However, depending on one's modeling purpose, the increased model complexity may not be beneficial, for example because it hinders understanding of model behaviour. Hence we develop a simplification

in the form of a climatology of aerosol concentrations. In one approach, the climatology prescribes properties important for cloud droplet and ice crystal formation, the gateways for aerosols to enter the model cloud microphysics scheme. Another approach prescribes aerosol mass and number concentrations in general. Both climatologies are derived from full ECHAM-HAM simulations and can serve to replace the HAM aerosol module and thus drastically simplify the aerosol treatment. The first simplification reduces computational model time by roughly 65%. However, the naive mean climatological treatment

needs improvement to give results that are satisfyingly close to the full model. We find that mean CCN concentrations yield an underestimation of CDNC in the Southern Ocean, which we can reduce by allowing only CCN at cloud base (which have experienced hygroscopic growth in these conditions) to enter the climatology. This highlights the value of the simplification approach in pointing to unexpected model behaviour and providing a new perspective for its study and model development.

## 1  Introduction

Climate models are used both to understand the Earth system and to project its changing behaviour. In building models, their representativeness and realism are taken to be important indicators of model quality. In this view, the models' scope has historically expanded (Edwards, 2011) to include various Earth system compartments and components, from land surface properties to atmospheric chemistry. Aerosol particles are one such Earth system component that has started to be represented

in climate models since the late 1990s (Brief, 2018). Aerosol particles are liquid or solid particles suspended in air, ranging from black carbon to sea salt or bacteria. They are important for the climate system, both with direct effects, such as by absorbing or scattering radiation, and indirect effects via their interaction with clouds (Lohmann and Feichter, 2005; Storelvmo, 2017).



In particular, aerosols serve as cloud condensation nuclei (CCN) and ice nucleating particles (INPs) and thereby facilitate water phase changes in the atmosphere. Small droplets have a high curvature, which increases their saturation vapor pressure

(Kelvin effect). Such high supersaturations are not reached in the atmosphere. Hence, cloud droplets do not nucleate homogeneously. Instead, cloud droplets nucleate on CCN. A hygroscopic aerosol particle takes up water when exposed to humid air (hygroscopic growth). As it grows, the CCN dissolves in the forming solution droplet and thereby acts as a solvent to lower the droplets' saturation vapour pressure (Raoult effect). Köhler theory combines the Kelvin and Raoult effect, which results in an equilibrium saturation pressure curve with a maximum at the so-called activation radius (see Fig. 3). Once a CCN reaches

this supersaturation and grows beyond the activation radius, it will continue to grow even with decreased supersaturation, and is hence termed an activated cloud droplet.

Similarly, the energy barrier associated with freezing is too high for cloud droplets to freeze homogeneously in the atmosphere. Until $-35\,^\circ$C cloud droplets freeze only heterogeneously on an INP, which serves to lower the energy barrier associated with the freezing process (Murray et al., 2012; Lohmann et al., 2016; Kanji et al., 2017). At lower temperatures, cloud droplets

freeze homogeneously, without the aid of an INP. The salts dissolved in smaller solution droplets lead to a freezing point depression. Thus, solution droplets freeze homogeneously only at low temperatures or high supersaturations with respect to ice. Alternatively, they can freeze heterogeneously with the aid of an INP (Lohmann et al., 2016).

Thus aerosols influence cloud properties. For example, when aerosol particle concentrations are higher, more cloud droplets form. Given the same amount of liquid water in a cloud, they have a smaller size. This delays precipitation formation, increasing

cloud lifetime (Albrecht effect, Albrecht (1989); Storelvmo (2017)). However, clouds also influence aerosol particle concentrations, e.g. with precipitation removing aerosol from the atmosphere via wet scavenging. These aerosol-cloud interactions are numerous, challenging to quantify and thus their resulting forcing is associated with a large uncertainty (Boucher et al., 2013; Bellouin et al., 2020; Bender, 2020). This is because the scales involved range from the interaction of micrometer particles to effects on the global energy balance. Observations to quantify these interactions at a global scale are inherently difficult (Quaas

et al., 2020). For climate modeling, the small scales involved require that the aerosol and cloud microphysical processes are parameterized. These parameterizations are inherently associated with underconstrained degrees of freedom and uncertainty. For example, an intercomparison conducted by Fanourgakis et al. (2019) shows that there is substantial disagreement in the CCN concentrations simulated by different global models.

The climate modeling community has responded to the challenges of aerosol and cloud microphysics (CMP) research by

expanding their models to account for an increasing variety of processes and compounds. This approach is grounded in the reductionist idea that a complex system can be decomposed into its parts, which then all need to be represented (Shackley et al., 1998; Saltelli et al., 2020b). While this makes the model more representative, its increasing complexity can arguably be counterproductive for the interpretability of the model (see Proske et al. (2023a) and Proske (2023)).

Thus the representative complexity paradigm in environmental modeling has come under challenge. For example, Cox et al.

(2006) developed a systematic approach to identify excess process complexity. By automatically replacing variables with constants in the code, they generated many simplified model variants. Comparing their results, they identified redundancies or overparameterizations. This approach has been successfully performed for example on wheat and soil models (Cox et al., 2006;



Crout et al., 2009; 2014). Working with the ECHAM-HAM CMP module, Proske et al. (2022) have introduced an approach that varies process efficiencies to test the models' sensitivity to these processes and thereby identifies potential for simplification.

Proske et al. (2023a) applied this method to 15 out of 17 processes in the CMP module and were able to simplify 7 of them.

For aerosol modules, various process sensitivity studies have been reported, but without a direct tie to model simplifications (see e.g. Schutgens and Stier (2014) for an extensive aerosol pathway analysis). Instead, aerosol module simplifications tend to be more drastic. For example, Liu et al. (2012) and Ghan et al. (2012) reduced the number of modes in their aerosol module MAM (part of CAM5) from 7 to 3 modes while still achieving satisfying performance in model results. Similarly, Zhu

et al. (2022) developed a parameterization of dry effective aerosol radius based on the mass of two species. This allowed them to use only a one moment (mass) prognostic representation and deduce the zeroth moment (number) at negligible computational expense. Furthermore, Ghan et al. (2013) developed a one dimensional model with physical aerosol and cloud processes included to aid in the exploration of how parameter uncertainty travels through to ACI uncertainty. Stevens et al. (2017) developed a plume climatology of anthropogenic aerosol based on the model- and observation derived MACv2 climatology. Their

climatological representation MACv2-SP was analytically based and consisted of only 9 plumes, but gave good agreements to the full MACv2 climatology. They highlight its use case in comparing aerosol responses in different models. Also, Fiedler et al. (2019) applied this fast MACv2-SP climatology in a scenario investigation. Presently, Weiss et al. (2023) are developing a simplified version of the HAM module, called HAMlite, by reducing aerosol tracers and imposing fixed aerosol composition. Number concentrations are still computed prognostically, but the predefined species composition allows to pre-compute

radiative properties. This allows the scheme to be used in convection-resolving model simulations.

The process of model simplification may seem counter-intuitive at first: we are used to taking model expanse and complexity as a sign for model quality. Yet any simplification sacrifices the representative depth of a model. However, the representativeness of a given model is but one goal of model development. Other goals include predictive capability and the generation of understanding. The concept of modeling visions is helpful to disentangle these goals (see Fig. 1). Melsen (2022) has termed the

modeling motivation and philosophy of any given modeler his or her modeling vision. We identify three climate modeling visions from science studies literature, which we term here the representative, predictive and heuristic modeling vision (Shackley et al., 1999; Shackley, 2001; Sundberg, 2009).

– Researchers with the **"representative" vision** (Sundberg, 2009) aim to represent the climate system in its full complexity, thus for them a more complex model will have a greater truth-content (Shackley et al., 1998; Shackley, 2001).

The ultimate goal would be a model that includes all processes and interactions in a realistic fashion (Schneider and Dickinson, 1974; Shackley et al., 1998; Parker, 2003).

– Modelers adhering to the **heuristic vision** aim for an understanding and exploration of the climate system. Thus, which model is 'state of the art' depends on the question that is being asked. Also, heuristic modelers may pursue modeling purely to advance and help their thoughts. They may see mathematical models' use in exploring questions, not in

asserting answers (Saltelli et al., 2020a).



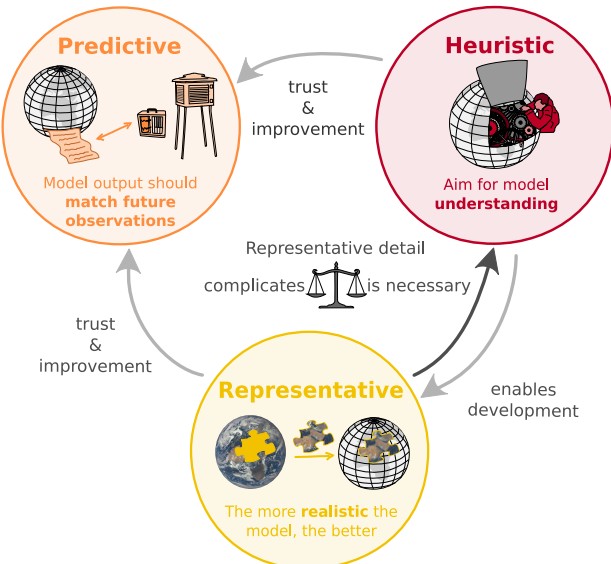

**Figure 1.** Sketch illustrating the different modeling visions and some of their relations. Following one vision may bring progress in another. The conflict between the representative and heuristic vision, that increased representative complexity makes models less interpretable, is what we aim to address in our work of simplification. Note that in the predictive vision, the match of model output to observations may also concern past observations, for example in hindcasts. Credits: see acknowledgements section.

- In the **predictive vision** (Sundberg, 2009), improving the forecast or the performance of the simulation with respect to observational data is the main goal, which leads to choosing practical over the best theoretical solutions.

Of course, the distinction between these visions is not sharp (Heymann and Hundebol, 2017). In practice, many modelers will find themselves sharing thoughts with all of these visions. In addition, as Fig. 1 points out, following one of the visions may

bring benefit to another. For example, when one learns about model behaviour, this may help to identify processes that need to be improved in order to gain predictive accuracy. However, the visions may also indicate conflicting avenues for model development goals. On the one hand, model complexity is encompassed by the representative vision. On the other hand, it makes models and their results more difficult to interpret and thus it harms the heuristic vision of model use for generating understanding. This conflict is what we aim to address with our simplification work.

One might object that simplifications may appear harmful even for the model's ability to supply understanding. For example, if one is interested in the process of aerosol particle coagulation, a model using a CCN climatology would not seem to be of much use. However, the simplification of one model part allows for easier investigation of other model parts, in our example e.g. cloud droplet coagulation. Also, a simplification of the process under study may help understanding, e.g. in identifying which factors influence a process and by enabling clearer sensitivity studies. When unsatisfying simplification results guide

the scientific exploration, the model itself is pointing the developer towards the important processes or behaviour that shape





the model response. Thus, attempts at drastic simplifications may open up new perspectives on the model understanding and development problem.

We develop a climatology of a) CCN that serves as the connection between the aerosol particles and the CMP and b) aerosol mass and number concentrations. Both climatologies in combination with a pre-existing climatology of aerosol radiative effects
can replace the aerosol module HAM. Such a top-down approach, of investigating how the model reacts to changes in aerosols, may be more effective than a tedious bottom-up approach of elaborating all possible processes, as Stevens et al. (2017) argue as well. Using the climatology allows to isolate remaining processes and their effects and study associated uncertainties. The climatology we develop and test is based on full-HAM model output. We demonstrate that this approach works in principle and discuss which features are required in the climatology. This gauges the possibilities for the use of an observation-based
climatology in the future, which would satisfy both the representative and the heuristic vision while avoiding representative complexity. Most importantly, as we demonstrate, the development of the climatology already opens up new avenues for understanding.

Our approach departs from the representative approach that has dominated Earth system model development and fueled an ever increasing model complexity. To be clear, our simplifications do not attempt to judge any process as unimportant
in reality. Rather, the model is our object of study. Where our findings deviate from physical understanding, this difference needs to be investigated and offers an avenue for model development. While our simplifications sacrifice representativeness for interpretability, the development for more comprehensive models may continue alongside, creating a model family of various complexity, where one may choose a configuration based on a given study's purpose. At the same time, our approach also satisfies the predictive vision of model development. In our simplification attempts we strive for equifinality (Beven, 2006;
Beven and Freer, 2001), meaning that the simplified model produces results similar enough to the full model for the purpose at hand, thereby providing equal predictive quality. Because the CCN climatology allows to replace the whole aerosol module HAM, it incurs large reductions in the model's run time. This may be used to save costs, run more or longer simulations, detail other processes, or increase the model's resolution.

The approach and results of this study are outlined in Fig. 2. The following section describes the aerosol climate model
ECHAM-HAM, the process implementations relevant to this study and the two CCN/aerosol climatology implementations (Sec. 2). In the presentation and discussion of results in Sec. 3, we describe the effects of both climatologies. We also detail the investigative process that co-evolved with the development of the climatology, highlighting how the approach of simplification generates model understanding. Section 4 discusses this approach and points out possible use cases of the CCN climatology.

## 2 Methods

This study employs the aerosol-climate model ECHAM6.3-HAM2.3 (Neubauer et al. (2019); Tegen et al. (2019), ECHAM-HAM hereafter), in the same configuration as in Proske et al. (2022). Its aerosol module HAM was implemented by Stier et al. (2005) (updated to HAM2 by Zhang et al. (2012)) using the 7 mode aerosol module M7 from Vignati et al. (2004). Aerosols of varying composition are grouped into 7 lognormal size distribution modes, which are distinguished by size and solubility



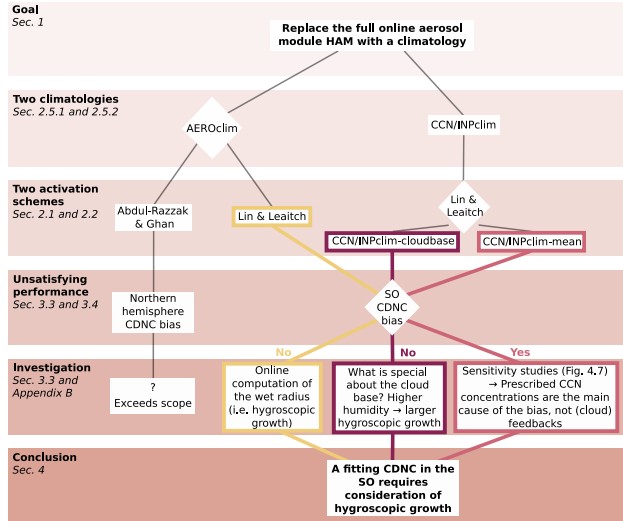

**Figure 2.** Flowchart illustrating our investigation of the different climatology versions.

(see Fig. 3). Various process treatments are included in HAM, for example condensation or coagulation moving particles
between modes. Hygroscopic growth of aerosol particles is implemented using Köhler theory with a prescribed hygroscopicity
parameter for each substance, following Petters and Kreidenweis (2007) (Zhang et al., 2012). The 2-moment CMP scheme
prognostically computes ice crystal and cloud droplet mass and number and diagnoses rain and snow mass concentrations
(Lohmann et al., 2007). For a detailed description of the included cloud microphysical processes see Proske et al. (2023a). As
in the real atmosphere, in ECHAM-HAM the aerosols influence CMP by serving as CCN or INPs in cloud droplet activation
and ice crystal nucleation. There are two cloud droplet activation parameterizations implemented into ECHAM-HAM (see
Table A1 for their separate tunings).

## 2.1 Lin & Leaitch cloud droplet activation

The cloud droplet activation following Lin and Leaitch (1997) (**act-ll** in the following) was implemented into ECHAM-HAM
by Lohmann et al. (2007). It empirically relates the number of nucleated cloud droplets, $\text{CDNC}_{\text{act}}$, to the aerosol number
concentration and updraft:

$$\text{CDNC}_{\text{act}} = 1 \cdot 10^9 \cdot \left( \frac{N_{\text{a}} \cdot w}{w + 2.3 \cdot 10^{-10} \, \text{m}^4\text{s}^{-1} \cdot N_{\text{a}}} \right)^{1.27} \tag{1}$$

Here, for $N_{\text{a}}$ ECHAM uses the number concentration of aerosols particles with wet radii $> 0.03 \, \mu\text{m}$[1]. Thus $N_{\text{a}}$ first needs to
be derived from the soluble aerosol size distributions in HAM (see Fig. 3). The updraft $w$ is calculated from the mean updraft,

---

[1]Note that this is different from the $0.035 \, \mu\text{m}$ cut-off radius that is stated in Lohmann et al. (2007), which Lohmann et al. (2008) changed to $0.03 \, \mu\text{m}$
to accommodate tuning constraints. This cutoff refers to stratiform clouds. For detrained convective clouds, the cutoff is radii $> 0.02 \, \mu\text{m}$ (introduced as
$0.025 \, \mu\text{m}$ in Lohmann (2008)).





*(Caption on next page.)*





**Figure 3.** Illustration of the points where aerosols influence cloud microphysics in the model and where hence the AEROclim and CCN/INPclim approaches are applied. Both climatologies are derived from a full HAM default simulation, which diagnoses monthly mean tracer diagnostics or potential CCN (and the other variables indicated in the illustration), respectively. Note that CCN/INPclim can be used only with the Lin & Leaitch (*act-ll*) and not with the Abdul-Razzak & Ghan activation scheme (*act-arg*). $\mathrm{frac_{dust/bc,act}}$ is the fraction of dust/black carbon in activated aerosol, $\mathrm{frac_{dust,ins}}$ is the fraction of dust in the insoluble coarse and accumulation mode, $\mathrm{r_{dust,wet}}$ is the wet radius of dust, $\mathrm{n_{sol}}$ is the number concentration of soluble aerosol and $\mathrm{r_{sol,wet}}$ is its wet radius. The CCN and INP climatology plots are illustrative, for one month and level (the climatologies really are four dimensional, resolved in space and time (in the form of monthly means)).

the turbulent kinetic energy and contributions from the convective available potential energy (see Lohmann et al. (2007) for
details).

Equation 1 is an empirical relationship derived in Lin and Leaitch (1997). They used aerosol and cloud droplet measurements from a field study in the North Atlantic[2] to evaluate two cloud droplet activation parameterizations, of Ghan (Ghan et al., 1993, 1995) and Abdul-Razzak (Abdul-Razzak et al., 1998). Both parameterizations were found to underestimate measured CDNC, but if one used the maximal updraft as input, both were able to predict the maximum measured CDNC well. Thus
Lin and Leaitch (1997) proposed to use these parameterizations to calculate the maximum CDNC and gave an empirical relationship to compute the mean CDNC. In their ECHAM-HAM implementation, Lohmann et al. (2007) used this empirical relationship in combination with the Ghan parameterization because the Abdul-Razzak formulation relied on supersaturations which may be unrealistic at the model grid scale (personal communication Ulrike Lohmann).

## 2.2   Abdul-Razzak & Ghan cloud droplet activation

The Abdul-Razzak & Ghan cloud droplet activation parameterization (**act-arg** in the following, introduced in Stier (2016) and Lohmann and Neubauer (2018), see Tegen et al. (2019)) is explicitly based on Köhler theory. In HAM, each of the seven modes has a different composition. Thus, Köhler coefficients need to be computed for each soluble mode separately (using Abdul-Razzak et al. (1998, Eq. 5) and Abdul-Razzak and Ghan (2000, Eq. 3 and 4); the nucleation mode is excluded from activation). From these coefficients and the updraft velocity, a maximum supersaturation is calculated. This is translated to an activation
radius for each mode via the mode radius and its corresponding critical supersaturation (Abdul-Razzak and Ghan, 2000, Eq. 12). Subsequently, all aerosols larger than the critical radius are activated into potential activated cloud droplets ($\mathrm{CDNC_{act}}$) for each mode.

---

[2]The field study was the North Atlantic Regional Experiment in 1993. Lin and Leaitch (1997) used data from 14 flights in and around stratus clouds over the Bay of Fundy and the Gulf of Maine in August and September. This illustrates the mismatch in complexity between the 28 tracer HAM module and the 15 processes in the CMP module that are connected through a parameterization that is based on data sampled in a constrained timeframe and location. The $2.3 \cdot 10^{-10}\,\mathrm{m^4 s^{-1}}$ factor is empirically based and was transmitted via personal communication from Richard Leaitch to Ulrike Lohmann.



## 2.3 Activated cloud droplets enter the cloud microphysics

Whether the potentially activated cloud droplets calculated in one of the previous parameterizations actually produce new cloud
droplets is determined in the CMP module. For all cloud bases at liquid or mixed-phase cloud conditions, it evaluates whether
$\mathrm{CDNC_{act}}$ exceeds the present CDNC. If this is the case, the CDNC at cloud base and throughout all cloud levels above is set
to $\mathrm{CDNC_{act}}$ that was calculated for the cloud base. This approach resembles the adiabatic ascent of an air parcel with cloud
droplet activation occuring mainly at cloud base.

## 2.4 Ice crystal nucleation in ECHAM-HAM

In ECHAM-HAM, ice nucleation mechanisms are parameterized as follows:

– **Heterogeneous freezing at mixed-phase temperatures** ($0\,^\circ\mathrm{C} > T > -35\,^\circ\mathrm{C}$) was introduced in Lohmann and Diehl
(2006) and distinguishes two types of heterogeneous freezing, which are considered to be important in the atmosphere:
Dust from the coarse and accumulation insoluble modes aids in **contact nucleation** (Lohmann and Diehl (2006); Hoose
et al. (2008), following Young (1974); Cotton et al. (1986); Levkov et al. (1995)). Its number concentration is multiplied
with an efficiency factor that accounts for the species specific temperature dependent INP efficiency. For this factor, the
dust is assumed to be composed of montmorillonite (values given in Hoose et al. (2008, Table 1)). Both mineral dust and
hydrophilic black carbon participate in **immersion freezing** (Lohmann and Diehl (2006); Hoose et al. (2008) following
Diehl and Wurzler (2004)). For their number concentration, the soluble accumulation and coarse mode of both species
are used (Lohmann and Neubauer, 2018). In practice, the fraction derived from dividing their activated concentration
by the activated CCN number concentration is used in the parameterization. Again, this is multiplied by an efficiency
to mirror the temperature dependent INP ability of the individual species. Again, for the mineral dust, montmorillonite
composition is assumed.

– **Homogeneous freezing of cloud droplets at cirrus temperatures** ($T < -35\,^\circ\mathrm{C}$) is realised simply by converting all
clouds droplets to ice crystals at these temperatures.

– **Homogeneous freezing of solution droplets at cirrus temperatures** ($T < -35\,^\circ\mathrm{C}$) was introduced in Kärcher and
Lohmann (2002) and Lohmann and Kärcher (2002). It uses the soluble aerosol number concentration and radius from
HAM as input. Since homogeneous ice nucleation may take place at high supersaturations with respect to ice, it solves
for the competition between updraft creating supersaturation and crystal growth depleting it. Heterogeneous freezing of
solution droplets at cirrus temperatures ($T < -35\,^\circ\mathrm{C}$) has been implemented into ECHAM-HAM previously (Lohmann
et al., 2008), but is not used in this study.





## 2.5 The aerosol/CCN climatology

### 2.5.1 CCN/INP climatology (CCN/INPclim)

To create the aerosol cloud climatology that enters *act-ll*, the number of potential CCN is diagnosed from a full HAM simulation. In this way, only one CCN concentration instead of all HAM tracers needs to be saved, while the dependence of cloud
droplet activation on the updraft velocity is still considered (see Fig. 3). This total number is thus independent of the aerosol composition. Apart from the CCN concentration, the soluble aerosol particle concentrations (without a cutoff radius) as well as their mode radii are supplied to the homogeneous freezing of solution droplets. The particle concentration is determined from the potentially available CCN that is also used as input to *act-ll*. This introduces a deviation from the default setup, where the entire Aitken, accumulation and coarse soluble modes are used, without any lower limit for their radii. In processing the output
from the default setup, the maximum of the monthly mean CCN over all conditions and those at cloud base was used to create the monthly mean, 3D resolved climatology (CCN/INPclim-cloudbase), i.e.

$$\mathrm{MAX}(\mathrm{MEAN}(\mathrm{CCN}_{\mathrm{all}}), \mathrm{MEAN}(\mathrm{CCN}_{\mathrm{at\,cloudbase}})).$$

This is because the cloud base condition (which for cloud droplet activation is restricted to $T > -35\,^{\circ}\mathrm{C}$) leads to an underestimation of CCN at cirrus temperatures and thus for the homogeneous freezing of solution droplets. Hence our climatology uses
the maximum of the mean CCN concentration over all conditions and the mean cloud base CCN concentration.

Since the CMP scheme uses CCN only at cloud base for activation, CCN/INPclim-cloudbase represents cloud droplet activation conditions in the model more specifically. The cloud base condition is the relevant one because in the current implementation activation is limited to cloud base and the enhanced CDNC is taken to be the same at higher cloud levels (see Fig. 3). Using only the mean CCN concentration ($\mathrm{CCN}_{\mathrm{all}}$) was also tested (CCN/INPclim-mean). In fact, investigating the difference
between CCN/INPclim-mean and -cloudbase allowed us to formulate the more appropriate conditions for a CCN climatology (see Sec. 3.3).

The treatment of heterogeneous contact freezing of CDs requires the fraction of insoluble dust aerosol as well as its radius as input. They are also diagnosed from the full HAM simulation. For immersion freezing, the fraction of dust and black carbon aerosol of the activated aerosol is used. These quantities are used as 3D monthly mean fields in the climatology.

### 2.5.2 Aerosol climatology (AEROclim)

*Act-arg* requires detailed composition as well as number concentrations for each mode. Hence, to be able to use the parameterization in its present form, all these properties needed to be prescribed and using the CCN/INPclim approach does not work for *act-arg*. In the implementation, we opted for a pragmatic approach: in the default simulation, all aerosol tracers (mass and number for each mode and species) were diagnosed as monthly means. In the climatology simulations these were prescribed
and all processes that would change the tracers' concentrations were deactivated. AEROclim can of course also be used for *act-ll*.





Note that AEROclim supplies monthly mean aerosol concentrations, which are used to compute aerosol properties online. In the case of non-linear computations, the quantities computed from monthly means will not be equal to the monthly mean of those quantities. Thus one can expect deviations between CCN/INPclim (which uses mean quantities) and AEROclim (which

computes these quantities from the mean).

### 2.5.3   Aerosol radiation climatology

Aerosol particles' effects on climate are not limited to clouds, but they also exhibit a radiative effect on their own. Thus, when we replace HAM with CCN/INPclim, the radiative aerosol effect requires a special treatment as well. Different versions of a climatological treatment of aerosol radiative properties have been implemented into ECHAM-HAM already. For AERO-

clim, radiative properties can be computed online from the supplied aerosol concentrations. For CCN/INPclim, we use the Max Planck Institute Aerosol Climatology (MAC-v1) developed by Kinne et al. (2013), which is based on both photometer observations and model data (ARclim). Alternatively, for sensitivity tests we also use a setup with no aerosol radiative effect (NoAR).

### 2.6   Historical and future simulations

To test whether the simplifications that we derive from present-day (PD) simulations (run for 2003 or 2003-2012) impede the model's ability to simulate different climate states, we perform sensitivity simulations with each of the simplifications in pre-industrial (PI) and possible future conditions. Again we keep the applied condition changes simple: for PI simulations we merely use PI aerosol emissions (ACCMIP data (Lamarque et al., 2010), from 1850, 1870, or 1900 as data was available), but everything else (greenhouse gas concentration, SST and sea ice) for the year 2003. Similarly, to represent a warmed climate

state, we add a spatially resolved increase of SSTs that amounts to $4\,\mathrm{K}$ in the ice-free ocean mean (as in the AMIP-future4K experiments, see Webb et al. (2017)) to allow for cloud responses to warming. Again the simulation setup was left unchanged apart from the prescribed SSTs and simulations were run for the year 2003.

## 3   Results and discussion

Applying CCN/INPclim to replace HAM yields surprisingly equifinal results to a full ECHAM-HAMMOZ simulation. Fig-

ure 5 shows that the climatology causes some positive deviations in cloud droplet number concentration (CDNC) between $-30\,°\mathrm{N}$ and $30\,°\mathrm{N}$ in the liquid cloud regime, as well as at around $50\,°\mathrm{N}$ in the mixed-phase cloud regime. At $-50\,°\mathrm{N}$, CDNC is underestimated by CCN/INPclim-cloudbase in both temperature regimes. While these deviations are larger than the inter-annual variation of the default simulation, they are small in relative terms, remaining roughly $< 25\%$. The deviations in CDNC translate to deviations in the liquid water path (LWP), which are even smaller in relative terms. In the ice phase,

CCN/INPclim-cloudbase causes significant yet small positive deviations in the mixed-phase ice crystal number concentration (ICNC), restricted to the northern hemisphere. In the cirrus regime, CCN/INPclim-cloudbase leads to increases in ICNC at roughly $10\,°\mathrm{N}$ and $10\,°\mathrm{S}$ of the equator. These changes in ICNC result from a combination of the CCN/INPclim-cloudbase





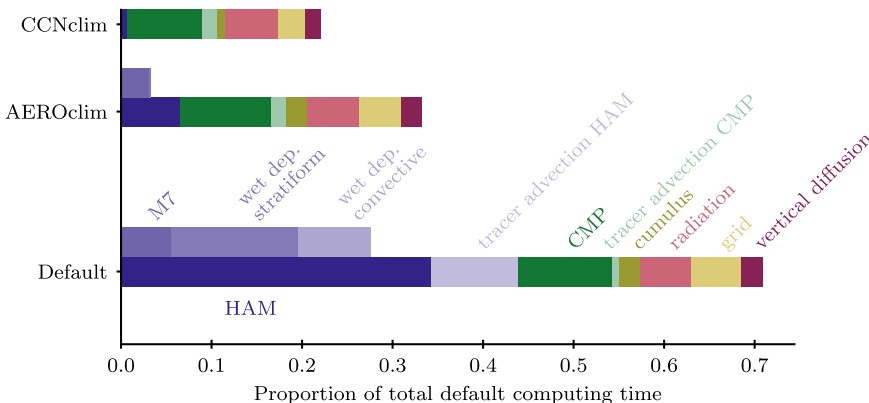

**Figure 4.** Relative computing time associated with the most expensive model parts, compared between a default and a CCN/INPclim model run (15 days, without writing output). The M7 aerosol microphysics are highlighted as part of the HAM aerosol module. Wet deposition is the most expensive single process and is called both in the stratiform and the convective cloud microphysics. Tracer advection handles both 28 aerosol and 2 cloud microphysics tracers and is divided here accordingly. In the case of CCN/INPclim and AEROclim the CMP tracer advection is not shared with HAM and thus the overhead is attributed solely to the CMP tracer advection timing, making it appear larger. *grid* names the transformations between the spectral and cartesian grid coordinates. For AEROclim, some HAM property computations are still executed, based on prescribed aerosol concentrations.

modifications in aerosols supplied to CCN activation, INPs and homogeneous freezing of solution droplets (see Sec. 3.2). The ice number changes do not translate to significant ice mass changes (see ice water path, IWP in Fig. 5), except for the positive deviation at roughly 10 °S. This in turn causes a positive deviation in longwave cloud radiative effect (LCRE).


Whether the deviations between the full aerosol scheme and the climatology are judged to be acceptable depends on one's modeling purpose. CCN/INPclim offers large savings in computational time. Figure 4 illustrates that HAM takes up more than 40 % of the computational time (excluding writing output). In comparison, the 2-moment CMP scheme takes up roughly 10 % and the transformations between the spectral and cartesian grid take up about 5 %. In AEROclim aerosol concentrations are

prescribed and excluded from advection, but still HAM needs to compute aerosol properties from the concentrations. Thus the simulation takes longer than CCN/INPclim. However, there is room for additional time savings: some properties like the wet radius are updated multiple times in the HAM routines, which may be reduced since processes affecting the wet radius are deactivated in AEROclim. In addition, some properties may be pre-computed with the frequency of the climatological input, similar to how Weiss et al. (2023) treat aerosol radiative processes. The single most expensive process related to aerosol

treatment is the wet deposition, which is called twice, in the stratiform and the convective CMP scheme. Thus, switching the wet deposition scheme from the more complex scheme of Croft et al. (2010) (size-dependent in-cloud and below-cloud scavenging) to a simpler scheme would enable large time savings of up to 20 %. Running the model with CCN/INPclim instead of HAM reduces the simulation time by 65 % (not shown, simulations for 15 months and including the writing of output with our standard requirements).



For predictive purposes, the large savings in computational expenses may outweigh the deviations in cloud particle concentrations. In fact, these may be minimized with further development of the climatology and tuning of the model, which we have not attempted. For our purpose of understanding model behaviour, CCN/INPclim-cloudbase results are deemed sufficiently similar to the default simulation to allow for a comparison. In particular, its development towards achieving such similarity opened up new perspectives (see Sec. 3.3).

## 3.1   Effect of the radiation climatology

Replacing HAM requires not only a new treatment of cloud active aerosols, but also replacing their radiative effects. In ECHAM, one can choose between interactive HAM aerosols (default), or climatologically prescribed aerosol radiative effects (ARclim, used in CCN/INPclim-cloudbase case), or no aerosol radiative effect (NoAR). To be able to judge the effect of the radiative climatology separately, Figure 5 compares the full HAM simulation with two that use HAM while treating only radiation climatologically. In terms of cloud variables, the ARclim and NoAR results remain within or close to inter-annual variability. Only for ICNC at mixed-phase temperatures, there is a large positive deviation at around $30\,°$N. The northern mid-latitudes are influenced by high concentrations of Saharan dust aerosol, which is radiatively active. The temperature changes induced by a change in the aerosol radiative climatology may influence heterogeneous freezing, because that is temperature dependent. Also, note that the ICNC at mixed-phase temperature is orders of magnitude smaller than the other hydrometeor concentrations, which are less affected by ARclim (in Fig. 5 for cold ICNC and both CDNCs the dark blue line remains within inter-annual variability). We thus do not expect the radiative treatment of aerosols to impact the performance of the aerosol cloud climatology. This is mostly the case: CCN/INPclim-cloudbase simulations with the climatologies derived from the three different radiative treatment simulations give only slightly different results for cloud properties. The most pronounced difference arises from using the CCN/INPclim-cloudbase climatology without radiative treatment of aerosols (light green in Fig. 5), which exhibits an increase in CDNC and LWP at around $30\,°$N and in turn increases the magnitude of the shortwave cloud radiative effect (SCRE). Also, in this simulation ICNC at cirrus temperatures is decreased at latitudes around $30\,°$N compared to the other CCN/INPclim-cloudbase simulations. Similarly as for heterogeneous freezing, the missing radiative effect of dust aerosols may induce local temperature perturbations and thus turbulent kinetic energy changes, to which the homogeneous freezing of solution droplets is particularly sensitive (Kuebbeler et al., 2012). In the CCN/INPclim-cloudbase configuration, the model may be deprived of some regulating feedbacks to these temperature changes, leading to larger differences between the radiatively different variants of CCN/INPclim than in the default variants.

    To sum up, the radiative aerosol climatology does not yield much difference in zonal mean cloud variables compared to the full HAM, but the combination of CCN/INPclim-cloudbase with ARclim enhances the differences. Thus, getting a perfect climatology for both CMP and radiation requires developing both together. However, the climatological treatment of aerosol radiative effects does not affect the main observations in the CCN/INPclim-cloudbase case. This eases our interpretation of CCN/INPclim-cloudbase and allows for a fair evaluation of CCN/INPclim-cloudbase against the default.



**Figure 5.** Zonal annual means, highlighting the effect of using a climatology for radiation. Uneven rows show absolute values for default simulations, and even rows show the deviations to the default simulation. All simulations are for the year 2003, and the grey shading indicates the maximum deviation from the 2003 default simulation in annual means between 2003 and 2012. The CCN/INPclim-cloudbase was either derived from a full HAM simulation (CCN/INPclim-cloudbase ARclim3) or from default simulations that in turn used an aerosol radiative climatology (CCN/INPclim-cloudbase from default ARclim or NoAR), where then the same radiative aerosol treatment was employed for the CCN/INPclim simulation.





**Figure 6.** As Fig. 5, but for the CCN/INPclim-cloudbase sensitivity simulations where parts of the climatology were set to 0.

## 3.2 Separate effects of the CCN/INP climatology

To test and illustrate the effect of CCN/INPclim-cloudbase on the various ways in which aerosol particles seed cloud particles,
we conducted simulations in which each of the INP, CCN and solution droplet effects was set to 0 separately (for details on the
processes see Fig. 3).

Disabling heterogeneous freezing by setting INP concentrations to zero (*0 INP* in Fig. 6) decreases mixed-phase ICNC
and increases CDNC in mid-latitudes. This is particularly pronounced at $30\,°$N, where high concentrations of ice-nucleating
Saharan dust particles are prevalent. The confined and overall small effect of heterogeneous freezing of cloud droplets on
the hydrological cycle agrees with what previous studies have found for ECHAM (Hoose et al. (2008); Dietlicher (2018);
Dietlicher et al. (2019); Villanueva et al. (2021); Ickes et al. (2022, 2023); Chadzelek (2023); Proske et al. (2023a)). The



division by temperature regimes highlights that this effect is hidden in the global mean ICNC, since this quantity is dominated by cirrus ICNC.

Aerosol particle concentrations in the model are also used to derive solution droplet concentrations, where solution droplets freeze homogeneously in the cirrus regime. Accordingly, when we set these to zero in the climatology (*0 SD* in Fig. 6), cirrus
ICNC decrease dramatically. Other cloud particle concentrations are unaffected by this change.

On the contrary, supplying zero aerosol particle concentration for cloud droplet activation (*0 CCN* in Fig. 6) dramatically affects all mass and number particle concentrations, decreasing both water and ice. This test illustrates the fuctioning of our CCN/INPclim-cloudbase climatology. However, one may wonder why significant water and ice mass mixing ratios are retained even without any CCN. This highlights the role of the CDNC minimum value in ECHAM, which serves to enforce a minimal
number concentration in order to avoid unrealistically large cloud droplets. Note that the total water (sum of water vapour, liquid and ice water mixing ratios) is conserved and that condensation is calculated from a saturation adjustment approach, i.e. cloud liquid water is also formed in the absence of CCN. In clouds with liquid water content, that minimum CDNC is dynamically calculated in our setup (see Proske et al. (2022, Sec. 2.1)). Thus, the liquid water mass is distributed over a small number of cloud droplets instead of being reduced to zero.

### 3.3   CCN/INPclim development

The difference between the two approaches to generate CCN/INPclim (CCN/INPclim-mean and CCN/INPclim-cloudbase) illustrates how the development of simplifications may force us to question and update our model understanding (illustrated in Fig. 2). Figure 7 shows that the results of the two approaches deviate most for Southern Ocean (SO) CDNC. This is strongly underestimated when employing CCN/INPclim-mean. Representing liquid clouds in the SO is a particularly common challenge
for climate models in general (Bodas-Salcedo et al., 2016; Kay et al., 2016; McCluskey et al., 2023), but it is not obvious why the mean climatology would deviate so strongly from the default simulation.

Investigating this difference further, Figure B2 shows that indeed this deviation is restricted to the SO (by this term we mean latitudes between $-40\,°N$ and $-80\,°N$ in the following). The underlying potential CCN concentrations show an overestimation of CCN by CCN/INPclim-cloudbase in the SO, compared to the default. While both CCN and CDNC have strong seasonal
differences in the SO, the relative differences with CCN/INPclim-mean or -cloudbase do not. These results indicate that CDNC in the SO can only be achieved with a monthly CCN climatology that overestimates CCN concentrations relative to the mean, regardless of season or baseline concentration. It is clear that the activation of CCN into cloud droplets is a process where large concentrations are important, because these have the potential to raise CDNC (because activated CCN can only raise CDNC if their concentration is higher than the pre-exisitng CDNC). However, it is not a priori clear why the CCN difference is especially
large in the SO, leading to large differences in CDNC there. To understand this behaviour, we have conducted various sensitivity simulations that help to exclude some hypotheses (see Appendix B and Fig. B3). These sensitivity simulations helped us to exclude interactions with the CMP or other feedbacks and influences as the source of the SO bias. They clearly point to the prescribed CCN themselves being the reason for the CDNC discrepancy. In parallel, we had developed AEROclim, which prescribed aerosol rather than CCN concentrations (see Sec. 3.4). Using AEROclim with *act-ll* alleviated a SO discrepancy





**Figure 7.** Zonal annual means, highlighting the effect of the different climatology variants. Uneven rows show absolute values for default simulations, and even rows show the deviations to the default simulation. *AEROclim act-arg* is shown in terms of difference to the *default act-arg* simulation. All values are annual means for the years 2003-2012, and the grey shading indicates the maximum deviation from the 2003 default simulation in annual means between 2003 and 2012. The climatologies were derived from the 10 year default simulations.



similarly to CCN/INPclim-cloudbase. This reinforced the idea that the bias in the SO does not stem from the use of a monthly climatology per se, but that the prescribed CCN values must be at fault.

The key to the SO bias proved to be the question: "why do cloud base conditions lead to higher CCN concentrations"? In clouds, the humidity is larger than outside of clouds. For soluble aerosol particles, increased humidity implies hygroscopic growth. Whether aerosols have experienced hygroscopic growth is relevant for *act-ll*, because it relies on the wet aerosol radius to estimate the concentration of CCN (see Fig. 3). Indeed, Fig. B1 shows that diagnosing CCN over all conditions implies a smaller wet radius than diagnosing it only at cloud bases. Hence, diagnosed CCN concentrations are smaller. This effect is especially pronounced in the SO. This can be explained by the larger relative humidity over oceans in general, as well as by the aerosol species composition. As stated above, the SO aerosol is dominated by sulfate and sea salt, which are much more hygroscopic than other aerosol species such as mineral dust (Lohmann et al., 2016). In the AEROclim case, the change in wet radii due to hygroscopic growth and the size cut-off are performed in online calculations. Thus, online computed CCN concentrations take cloud base conditions into account. In sum, in order to use a CCN climatology for *act-ll*, hygroscopic growth of particles needs to be taken into account (as in AEROclim), or CCN need to be diagnosed at cloud base conditions already (as in CCN/INPclim-cloudbase).

### 3.4 AEROclim

AEROclim alleviates the SO bias, but instead, zonal mean CDNC and LWP values are rather overestimated with AEROclim. For *act-arg*, AEROclim overestimates CDNC by about a third in the Northern Hemisphere, between $25\,°\mathrm{N}$ and $75\,°\mathrm{N}$. This bias may be subjected to similar tests as the ones we have performed for the SO bias above to elucidate its underlying cause. In particular, one would need to investigate the error that is introduced with AEROclim, where aerosol quantities such as the radius are computed from monthly mean concentrations using nonlinear relations.

### 3.5 CCN climatologies in different climate states

Figure 8 illustrates the performance of the climatologically simplified model with respect to the full HAM default. The different climate states manifest themselves e.g. in a decrease in northern hemisphere CDNC and LWP for the pre-industrial simulation. Regardless of these differences in the default simulations, the change induced by the simplification is similar for these different climate states. The overestimation of CDNC and LWP in the northern hemisphere decreases in pre-industrial climate. We attribute this to the decrease in absolute numbers that is present in the default simulation, Also, the role of minimum CDNC is bound to be more prominent in PI conditions and the minimum condition is present in both the full HAM and simplified model version. In the predictive vision, simulations in varying climate states serve to infer the reaction of the climate system to such changes. Table 1 summarizes the aerosol radiative forcing (ARF) and climate sensitivity ($\lambda$) as two key quantities of interest in this regard (see e.g. Bellouin et al. (2020)). $\lambda$ is well preserved by the simplifications. ARF exhibits differences of up to $\approx 35\%$, which are still small compared for example to the probability distribution of ARF presented in Bellouin et al. (2020). This is despite a strong mismatch between default and simplified model in the radiation balance at the top of the atmosphere in all climate states (see Fig. 8). Since the CCN climatologies are a drastic simplification, some mismatch is to be expected.





**Figure 8.** As Fig. 7, but comparing the different variants of the aerosol/CCN climatology for different climatological states. There are three default simulations (present-day, pre-industrial and plus 4 K conditions, as described in Sec. 2.6) in the uneven rows. The differences in the even rows are with respect to the respective climatological default state. See Fig. C1 for the AEROclim simulations, and Fig. C2 for a direct comparison of the differences between climate states.



**Table 1.** Comparison between the simplification simulations in terms of effect on aerosol radiative forcing (ARF) and climate sensitivity, computed from 10-year means. ARF is the difference between present day (PD) and pre-industrial (PI) radiation balances ($F_{net}$ (PD) - $F_{net}$ (PI)). The climate sensitivity $\lambda = \frac{\Delta T_{sfc}}{-(F_{net,FUT}-F_{net,PD})}$ is computed from the change in sea surface temperature $\Delta T_{sfc}$, which is $4\,K$ in the future4K experiment (FUT). All other quantitites (first four columns) are given for the PD simulation. $F_{net}$ is the net top of the atmosphere radiation balance. Cloud droplet number concentration (CDNC) and liquid water path (LWP) are vertically integrated. The simulations were conducted as detailed in Sec. 2.6, with the default simulations from the respective climate state providing the climatological input to the sensitivity simulations (the 10-year mean of CCN/INclim or AEROclim was used). Fig. 8 and C1 and C2 in Appendix C show the simulation results in more detail.

| Simulation | $F_{net}$ $(Wm^{-2})$ | CDNC $(m^{-2})$ | LWP $(m^{-2})$ | SCRE $(Wm^{-2})$ | ARF $(Wm^{-2})$ | $\lambda$ $(Km^2W^{-1})$ |
|---|---|---|---|---|---|---|
| default act-ll | 0.57 | 2.5 | 82 | -49 | -1.7 | 0.65 |
| CCN/INclim-mean | 0.25 | 2.1 | 80 | -48 | -1.9 | 0.60 |
| CCN/INclim-cloudbase | -2.0 | 2.6 | 86 | -50 | -2.3 | 0.66 |
| AEROclim act-ll | -1.7 | 2.7 | 88 | -52 | -2.0 | 0.63 |
| default act-arg | 0.18 | 3.3 | 82 | -50 | -1.9 | 0.65 |
| AEROclim act-arg | -2.5 | 3.7 | 91 | -54 | -1.9 | 0.64 |

As Table 1 shows, this mismatch is only to some part due to changes in CDNC, LWP and subsequently SCRE. Comparing to Fig. 5, it becomes clear that the climatology of the aerosol radiative effect adds to the discrepancy in radiative balance as well.

However, the fact that the mismatch between full HAM and CCN/INPclim is consistent between climate states suggests that on the one hand, it could likely be alleviated by tuning the CCN/INPclim model or improving the radiation climatology. On the other hand, differences between climate states are not affected by the mismatch, and thus the ability of the model to serve in studies of climate change (sensitivity) is preserved in principle. Of course, if one's objective is to study the changes induced by changing aerosol concentrations in detail, a detailed aerosol model will probably suit that purpose better.

**4  Summary, conclusions and outlook**

We have simplified the aerosol module HAM by using it to generate a climatology. The climatology then serves as input to the interfaces from aerosols to CMP. We have developed two versions of the climatology, one that prescribes CCN and INP and can only be used for the empirical Lin & Leaitch cloud droplet activation scheme, and one that prescribes aerosol mass and number concentrations that can also be used with the Köhler theory based Abdul-Razzak & Ghan scheme. For both versions we could

develop a climatology that globally gives promising results for our purpose of studying clouds. Regarding the simplifications fitness' for simulations in a different climate, Fig. 8, C1 and C2 in Appendix C show that the use of the climatologies inflicts no structural error in different climate states. The differences induced by the simplifications in relation to the default are mostly within inter-annual variability of each other in different climates. At the same time, differences between default simulations for different climates are large, confirming that they can serve as test cases. Table 1 shows that the simplifications preserve





estimates of climate sensitivity, but exhibit differences in aerosol radiative forcing of up to $\approx 35\%$ (for CCN/INPclim). The fact that we do not achieve equifinality in present day conditions and aerosol forcing highlights that the full complexity of the aerosol scheme has merit in the sense that it is not fully replaceable in a naively simple way. However, the climatologies and the tuning of the results to the default model can certainly be improved (see the present day radiative forcing in Table 1). We did not tune the simplification variants on purpose, to facilitate a clear comparison. Thus, this work demonstrates that such

drastic simplifications of aerosol treatment are possible.

The simplifications result in large computing time savings of roughly 65%. This suggests the use of the climatologies in settings where computational time is limited as e.g. for long climate simulations, high resolution simulations or large ensembles. We have also claimed that simplifications can enhance the interpretability of a given model. An explicit aerosol treatment may be necessary for example for studies of their health impacts and air pollution. In our case, our interest lies in studying the

modeled clouds and their properties. Thus, for this purpose it is a strength of our simplifications that they allow us to isolate and investigate only the cloud response to aerosols and not the feedback response of aerosols.

In addition, we have gained knowledge on which features of such a climatology are important. In terms of CDNC, results similar to the default could only be obtained with climatologies that take into account hygroscopic growth at cloud base conditions, which implies higher CCN concentrations. The CCN/INPclim gives satisfying results only when using CCN

concentrations at cloud base to generate it. The mean CCN over all conditions at all times is smaller and results in a large underestimation of CDNC in the SO. With sensitivity experiments we have excluded various cloud feedbacks and related factors as reasons for this behaviour. This lead us to conclude that in the SO CCN vary with cloud base conditions. We can explain this with hygroscopic growth, which increases the wet radius of aerosols in more humid cloud conditions and hence leads to higher CCN concentrations at cloud base with the *act-ll* scheme. This finding has important implications for the general sim-

plification of using a CCN climatology. Either, one would need to take hygroscopic growth on top of the prescribed CCN (as AEROclim) into account, or use a climatology that is derived from cloudy conditions already (as CCN/INPclim-cloudbase). The difficulty we had in interpreting the CCN/INPclim-mean model behaviour points to another strength of the simplifications. While developing simplifications the models forces us to look at parts that are important to the model itself. Instead of being guided by our a priori believes (which in our case pointed towards time variability or precipitation feedbacks), simplifications

thus allow for a change of perspective that may provide fresh insights.

More aerosol climatologies for use in climate models have been developed previously. It is important to stress that our climatologies are not meant to be generally applicable. Rather, we propose the process of simplification as a way to gain a new perspective on model behaviour and the simplified model as an explorative tool for further study. This distinguishes our climatologies from the MACv2SP climatology developed by Stevens et al. (2017). Their climatology is analytical, which

enhances its flexibility, the clearness of its assumptions and the possibilities for porting it to different models. Our climatology is meant for use in ECHAM with the goal of making its model results equifinal to the default ECHAM-HAM configuration. As such, not the single realisation of our climatology is important, but we have rather developed the model code to easily derive and employ new climatologies. This allows the kind of sensitivity studies we used to investigate the SO behaviour, which can be helpful in adapting to new model versions and investigating their differences. Further, MACv2SP is restricted to



anthropogenic aerosols and prescribes a change to CDNC directly. We take into account all of HAM's (soluble) aerosols. We specifically prescribe either aerosol or potential CCN that enter the cloud droplet activation schemes online and thus keep e.g. the updraft dependence.

## 4.1 Outlook

Our climatologies are explorative and meant to aid understanding. However, the results are encouraging to the idea that for
the purpose of studying clouds, the full aerosol module HAM is replaceable with a climatology. This opens the door to use observationally developed climatologies in the same setup. In this way, representative complexity could be replaced with a representative climatology. In fact, an observation-based CCN climatology may be more representative than the full HAM model itself, as the latter is known to exclude e.g. aerosol species such as nitrate, whose effect would be present in observations. Such observation-based CCN climatologies already exist. For example, Choudhury and Tesche (2022) derived one from the
lidar on the satellite CALIPSO. Importantly, our study shows that the use of mean CCN climatologies will not suffice. An additional treatment of hygroscopic growth is needed, especially in the SO. To this end, one approach to be tested is to apply the difference between CCN/INclim-cloudbase and CCN/INclim-mean to scale and adapt an observation-based climatology for use in ECHAM-HAM.

Alternatively, one may modify CCN/INclim-cloudbase to tune it towards observations, to test ECHAM-HAM sensitivities
towards this more representative CCN climatology. Note that observation-based climatologies limit research to present day aerosol conditions. In addition, using them for *act-arg* would require more detailed information than potential CCN concentrations on the side of the climatologies. Other model-derived CCN climatologies (as e.g. from Costa-Surós et al. (2020)) could address these concerns and may be used where their origin is thought to be superior to ECHAM-HAM-derived ones in epistemic terms. Also, by using these climatologies as input for the *act-ll* configuration, one could further elucidate the sensitivity
of ECHAM towards CCN. Similarly, the simplified model version may be compared to the single moment CMP scheme that is available for ECHAM, where CDNC is prescribed, to spotlight the role of activation in the model.

Of course our developed climatologies can be improved upon or made more sophisticated. For example, allowing for wet scavenging with a relaxation back to the climatology as it was implemented by Costa-Surós et al. (2020) in their ICON Large Eddy Simulations, would enable a reaction of CCN concentrations to cloud behaviour. However, the climatology is engineered
to provide adequate results in CDNC, and not to give a best estimate of CCN concentrations (CCN/INPclim-mean would for example be more apt for at least providing the model's best estimate). Hence, in its present form, also a direct comparison of CCN/INPclim-cloudbase to observational CCN concentrations, for example to judge the model performance, would go against its purpose. Just as MACv2SP, our parameterization is meant to be a reference climatology for the effect of CCN on clouds, and "not a reference aerosol climatology" (Stevens et al., 2017).

Instead, the simplified modules allow for an easier comparison between models by eliminating differences in details. For example, cloud microphysical schemes may be compared more easily between two models using the same CCN climatology. Our implementation of the climatologies also enables easily devised sensitivity studies, for example using a different time resolution for the climatologies. In particular, AEROclim opens up many possibilities for sensitivity experiments, for example



by setting single prescribed variables to 0. Thus our approach also has potential for model development. It highlights what
variables or features are important for model performance, and can serve to detect unintended behaviour or mistakes in the
code. Importantly, it shows the benefit for understanding in simplification, calling into question the representative complexity
paradigm that has dominated climate model development.

*Code and data availability.* The ECHAM-HAMMOZ model is freely available to the scientific community under the HAMMOZ Soft-
ware License Agreement, which defines the conditions under which the model can be used. The specific version of the code used for this
study is archived in the ECHAM-HAMMOZ SVN repository at /root/echam6-hammoz/tags/papers/2023/Proske_et_al_2023_ACPD. More
information can be found on the HAMMOZ website (https://redmine.hammoz.ethz.ch/projects/hammoz, last access: 22 November 2023).
Analysis and plotting scripts are archived at https://doi.org/10.5281/zenodo.10171426 (Proske et al., 2023d). Generated data is archived at
https://doi.org/10.5281/zenodo.10184958 and https://doi.org/10.5281/zenodo.10183962 (Proske et al., 2023b, c).





## Appendix A: Tuning

**Table A1.** Tuning parameters for the act-ll scheme that differ with respect to the ECHAM-HAM model version used in Proske et al. (2022) and Proske et al. (2023a) (see their Tables A1). The latter's tuning is used for the act-arg scheme. $\gamma_r$ is the scaling factor for the stratiform rain formation rate by autoconversion. $\gamma_s$ is a scaling factor for the stratiform snow formation rate by autoconversion.

| Parameter | act-ll this study | act-arg this study |
|:---:|:---:|:---:|
| $\gamma_r$ | 3.25 | 5 |
| $\gamma_s$ | 900 | 600 |



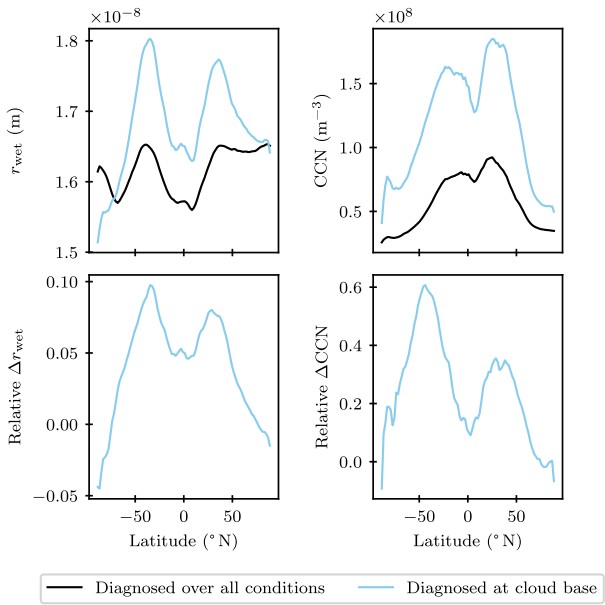

**Figure B1.** Zonal mean of vertically averaged wet radii and CCN, diagnosed from all conditions and only at cloud base. The second row shows the zonal mean of the vertically averaged differences.

## Appendix B:  Sensitivity simulations to elucidate the SO-bias

To understand the SO-bias in CDNC exhibited by CCN/INclim-mean (see Sec. 3.3), we conducted various sensitivity simulations:

– **Aerosol species emissions** – The SO aerosol composition is dominated by sea salt and sulfate, whereas in the Northern Hemisphere dust and black or brown carbon are more important. Sea salt and DMS emissions (with DMS being a precursor for sulfate) are highly dependent on wind speed, which might lead to high variability, making the climatological representation less suited for the SO. We tested this hypothesis by prescribing a fixed wind speed in the emission computations. This approach changes the default CDNC as expected, but the bias in the SO remains. As an extreme test, we turned off DMS and sea salt emissions separately (not shown) and together. This reduces CDNC both in the default and the CCN/INPclim-mean case. Turning off both DMS and SS, the difference between the full HAM and climatology simulation in the SO disappears, but CDNC are small in the full HAM simulation to begin with. Turning both emission types off separately does nothing to alleviate the SO bias.

– **Nudging** – By nudging pressure, vorticity and divergence we can test whether feedbacks from the cloud behaviour to the dynamics of the model are contributing to the discrepancy. Since the SO discrepancy remains in the nudged case, we can exclude these types of feedbacks as a reason for it.





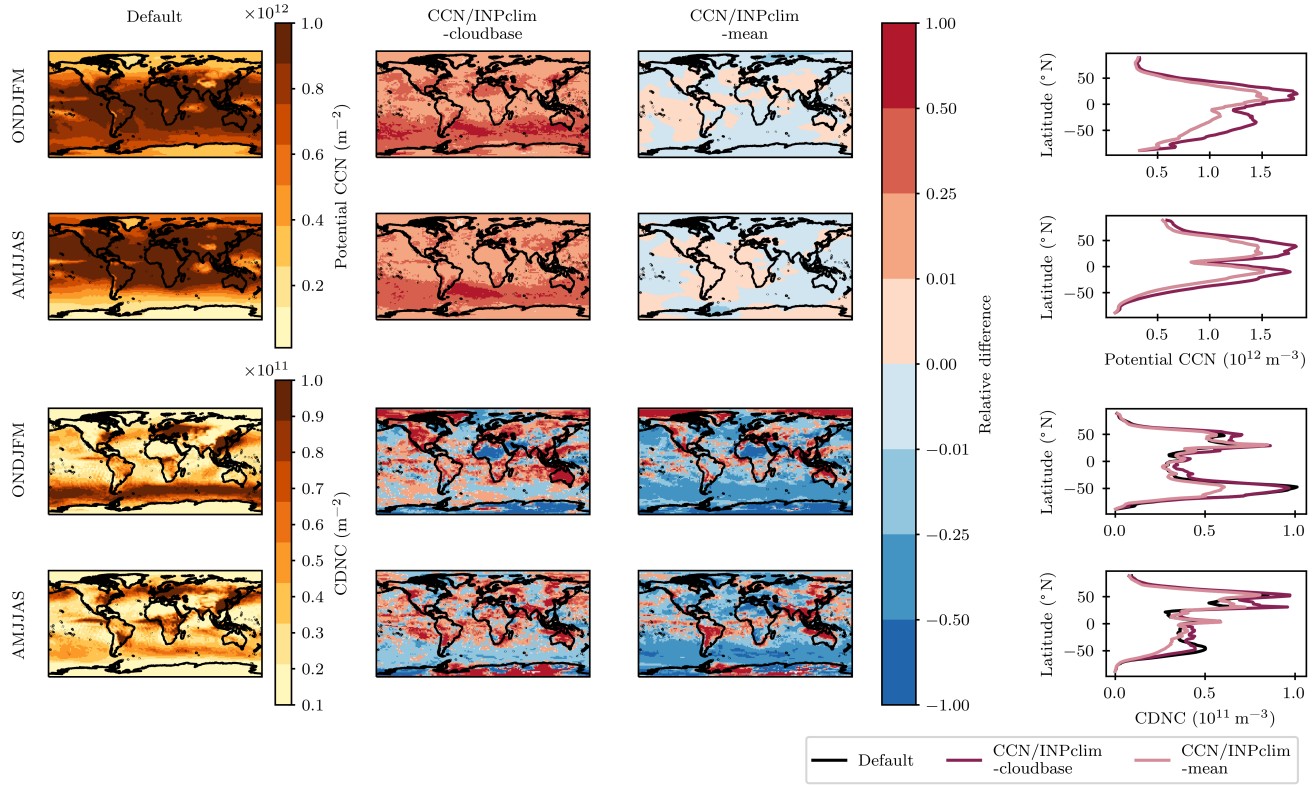

**Figure B2.** Seasonal mean maps of potential stratiform CCN concentrations (top two rows) and CDNC concentrations (bottom two rows), in absolute terms for the default simulation (left most column) and relative deviations for the different climatologies. The right most column shows the absolute zonal means. Note that for potential CCN, the default is hidden behind the CCN/INPclim-mean line.

– **CDNCmin** – To avoid unphysical situations, climate model code employs thresholds and other "minor-looking treatments" (Kawai et al., 2022). One of these is the CDNC minimum that serves to avoid situations where the model calculates a cloud with too few hence too large cloud droplets. In our ECHAM-HAM configuration, the minimum is calculated dynamically from the in-cloud water content and a set droplet radius (see Proske et al. (2022, Sec. 2.1)). CCN concentrations below the CDNC minimum threshold value are never effective in promoting cloud droplet formation.

However, they do enter into the CCN/INPclim climatology and may thus artificially lower effective values. We tested the effect of this threshold for the CCN climatology by allowing only CCN larger than the minimum to enter the climatology. Since the sensitivity simulation preserves the SO bias we can exclude CDNCmin as the reason for the SO discrepancy.

    – **Fixed updraft** – As illustrated in Fig. 3, both the potential CCN and the local updraft enter the calculation of activated cloud droplets. This updraft could be affected by dynamical feedbacks to a CCN perturbation. Thus we conducted simu-

lations where we put the value of the updraft that is used in the activation calculation to a constant value. This of course





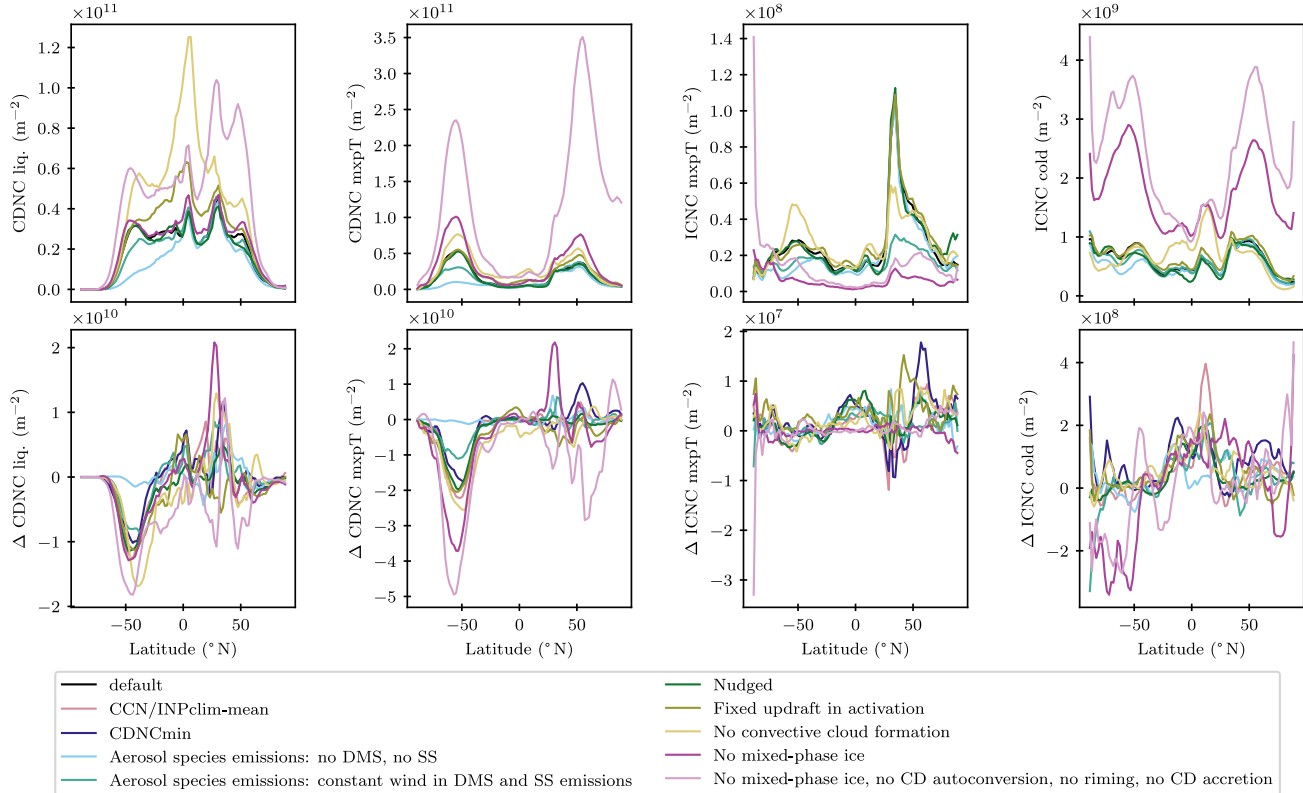

**Figure B3.** As Fig. 5, but for sensitivity simulations with different ways to create the CCN/INPclim-mean climatology. Differences are shown relative to the respective default simulations (indicated by the colors). Only the default mean climatology and the one created using only CCN values higher than the CDNC minimum are shown as differences to the default simulation (shown in black). Dimethyl sulfide (DMS) is a precursor of sulfate aerosols, SS stands for sea salt and CD for cloud droplet. For the "no mixed-phase ice" simulations, heterogeneous cloud droplet freezing as well as ice crystal sedimentation were inhibited.

deteriorates the performance in the default simulation, but since it does not reduce the SO bias from CCN/INPclim-mean, we can exclude the updraft hypothesis as well.

– **Convective cloud formation** – The CCN/INPclim prescribes CCN not only for stratiform but also for convective clouds. Cloud droplets that formed in convective clouds may enter stratiform clouds by detrainment. To test the effect of detrain-
ment we inhibited convective cloud formation. This of course changes the default simulation, but again it does little to reduce the SO CDNC bias.

– **Mixed-phase ice influence** – In the SO we expect different cloud phase distributions and cloud properties than in the Northern Hemisphere (Mülmenstädt et al., 2015), with e.g. lower ice crystal number concentrations in the SO (see Fig. 5). The difference in cloud phases could explain differing reactions of the clouds to changes in CCN and cloud





droplet formation. We tested this hypothesis in a simulation where all mixed-phase clouds were forced to remain liquid by inhibiting heterogeneous cloud droplet freezing as well as ice crystal sedimentation. The resulting simulations show neither a better CCN/INPclim-mean performance in the SO in terms of CDNC, nor a worse performance in the Northern Hemisphere. Hence we can exclude the ice phase as a reason for the SO discrepancy.

– **CMP processes** – Other CMP processes might lead to feedbacks that enhance the CDNC discrepancy in the SO. We
tested this hypothesis by turning off both the ice phase influence (as above), and the processes of cloud droplet auto-conversion, riming and cloud droplet accretion. Hence in this simulation all processes leading to liquid precipitation formation or influencing cloud droplet number concentrations (except nucleation) were inhibited. Removing the CDNC sink processes greatly enhances CDNC as expected. However, the underestimation of CDNC in the SO by CCN/INPclim-mean remains.



**Appendix C:  Simplification performance in different climate states (PD, PI and FUT simulations)**




**Figure C1.** As Fig. 8, but for the act-arg scheme default and AEROclim simulations.





**Figure C2.** As Fig. 8, but comparing the aerosol radiative forcing and difference between SST + 4 K and present conditions (as described in Sec. 2.6) between the default and CCN/INclim-cloudbase simulations.



*Author contributions.* UP developed the model code, ran the simulations, analysed the data, and wrote the manuscript. UP, SF, and UL developed the study idea and design and analysed the results. SF and UL edited the manuscript.

*Competing interests.* The authors declare no conflict of interest.

*Acknowledgements.* We would like to thank the people who discussed this work with us, providing ideas and valuable feedback: Philip
Stier, Philipp Weiss, and Peter Manshausen; the HAMMOZ 2023 workshop participants, especially Johannes Quaas; the participants of the clouds, aerosols, radiation and precipitation session at EGU2023, especially Anna Possner, Ali Hoshyaripour, Ann Kristin Naumann, Gabriella Wallentin, Luisa Ickes and Edward Gryspeerdt; and the EU project FORCeS participants. A special thanks to Franziska Glassmeier for providing constructive feedback on the manuscript. Fig. 1 used various files from wikimedia commons as templates for our sketch reproductions. The files are distributed under a creative commons attribution-share alike license or in public domain. We acknowledge the
authors Achodanick, Malchen53, NASA, Mrmw, Ninjastrikers, Agência Brasil, Jahobr and Ibex73. Throughout this study, the programming languages CDO Schulzweida (2018) and Python (Python Software Foundation, www.python.org) were used to handle data and analyse it. The visualisations have made ample use of Paul Tol's colour blind friendly colour schemes Tol (2021). The ECHAM-HAMMOZ model is developed by a consortium composed of ETH Zurich, Max Planck Institut für Meteorologie, Forschungszentrum Jülich, University of Oxford, the Finnish Meteorological Institute and the Leibniz Institute for Tropospheric Research, and managed by the Leibniz Institute for
Tropospheric Research (TROPOS). This project has received funding from the European Union's Horizon 2020 research and innovation programme under grant agreement No 821205 (FORCeS). This work was supported by a grant from the Swiss National Supercomputing Centre (CSCS) under project ID s1144. ECHAM-HAM simulations were also performed on the ETH Zürich Euler cluster.



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
