# Peer review of "Developing a climatological simplification of aerosols to enter the cloud microphysics of a global climate model"

_EGUsphere, 2023_

## Author Comment (AC1)

**Author Response to Reviews of**

**Developing a climatological simplification of aerosols to enter the cloud microphysics of a global climate model**

Ulrike Proske, Sylvaine Ferrachat, Ulrike Lohmann
*ACP,* `doi:10.5194/egusphere-2023-2783`
* * *
RC: *Reviewer Comment*,     AR: *Author Response*,     ☐ Manuscript text

We sincerely thank the reviewers for their constructive feedback. We implemented their feedback into a revised version of the manuscript. Please find our answers to the reviewers' points below, followed by a marked-up manuscript verison. Line numbers refer to the previous manuscript version.

In addition, we found a mistake in the calculation of the climate sensitivity $\lambda$. It was erroneously computed from the change in sea surface temperature. It is now corrected to use the actual global surface temperature change as diagnosed from the respective simulations (see Table 1). While this changed the overall result for $\lambda$, it did not increase the difference between the model versions.

> The climate sensitivity $\lambda = \frac{\Delta T_{\mathrm{sfc}}}{-(F_{\mathrm{net,FUT}} - F_{\mathrm{net,PD}})}$ is computed from the change in  surface temperature $\Delta T_{\mathrm{sfc}}$. All other  quantities (first four columns) are given for the PD simulation. $F_{\mathrm{net}}$ is the net top of the atmosphere radiation balance.

**1. Reviewer Comment #1**

RC: *I would like to provide some feedback on the authors' statement regarding the application of CCN/INPclim to replace HAM, resulting in "surprisingly equifinal results to a full ECHAM-HAMMOZ simulation." While I appreciate the study, I have concerns about the term "equifinal" given the observed relative variation of CDNC close to 20%, which, in my perspective, can be considered quite substantial. To ensure clarity, it would be beneficial to explicitly illustrate these relative differences, perhaps through the use of Figures 1-2.*

AR: *We appreciate this point and acknowledge that we have used the term "equifinal" too loosely. We have adapted the wording of the crucial passages to reflect more closely our strive for a simplified model version that resembles the default version (see quotes below). In that we are satisfied with a level of similarity that is adequate for our purpose of studying clouds. Specifically that means that deviations on the order of inter-model version differences are acceptable to us (see new Fig. B1 that we added in response to Reviewer 2). We were puzzled by your suggestion to use Figures 1 and 2 to illustrate the differences as we cannot see how one could do that, but we hope to have addressed your point with the changes in the text.*

The old

> Our climatology is meant for use in ECHAM with the goal of making its model results equifinal to the default ECHAM-HAM configuration.

now reads

> Our climatology is meant for use in ECHAM with the goal of making its model results resemble those of the default ECHAM-HAM configuration.

Note that the corresponding passage was moved in the text in response to Reviewer 2.

> In our simplification attempts we strive for  equifinality (Beven, 2006; Beven and Freer, 2001), meaning that the simplified model produces results similar enough to the full model for the purpose at hand, thereby providing equal predictive quality.

> Applying CCN/INPclim to replace HAM yields surprisingly  similar results to a full ECHAM-HAMMOZ simulation.

**RC:** *Moreover, the information presented in Figure B2 highlights relative differences in CDNC over the Southern Ocean ranging from -0.50 to -0.25. These variations are comparable to those induced by the replacement of microphysics schemes (e.g., replacing MG2 with P3). Consequently, it prompts the need for a thorough evaluation of whether the simplified aerosol treatment is scientifically justifiable.*

AR: *You are correct. We also judge these differences to be too large. However, please note that the large differences you are referring to occur with the rough simplification that we developed first. The simplification that we propose, our second development, reduces these biases tremendously, as can also be seen in Figure B2. The difference between the two, i.e. the quest for reducing the SO bias, is detailed in Sec. 3.3. On another note, in principle the replacement of a microphysics scheme should also keep results close to each other. In fact, we would expect the changes from the replacement of the whole aerosol scheme with a climatology to be larger than the changes induced by the switch between two different microphysics schemes, if both are scientifically justifiable.*

**RC:** *While the authors assert that a simple aerosol treatment will not significantly alter cloud properties such as cloud radiative forcing and LWP, it is imperative to ensure that the effective forcing of anthropogenic aerosols remains unaffected. Furthermore, I suggest paying close attention to the accuracy of the historical simulation of surface temperature anomalies. In essence, a more comprehensive effort is warranted to ascertain that the adoption of simple aerosol treatments does not have adverse effects on the estimation of aerosol radiative effects and historical simulations.*

AR: *Thank you for pointing out the need to be diligent in our assessment. We believe that in particular Sec. 3.5 addresses your concerns, as there we evaluate the performance in climate states with changed anthropogenic aerosols and sea surface temperatures, respectively.*

**RC:** *I appreciate your understanding and consideration of these points as the study progresses. Thank you for your diligence in addressing these concerns.*

AR: *Thank you for your comments, which gave us the chance to express the ideas contained in this manuscript more clearly.*

**2.  Reviewer Comment #2**

RC:  *Climatologies of aerosols for the ECHAM-HAM model are presented, motivated by the need for more interpretability. This is important and useful work and complements other efforts to simplify the aerosol microphysics such as HAMlite and MACv2-SP. An interesting analysis of the potential challenges of the simpler climatology approaches for CCN in humid conditions near cloud base is presented. The more complex aerosol climatology roughly maintains aerosol forcing and climate sensitivity as in the default model (at least, as it is calculated by the authors) while the simpler CCN/IN climatology does not maintain forcing: it's useful to see that some simplification can be achieved with minimal loss of 'equifinality' while more simplification, while still useful in some situations, does degrade key performance metrics. The paper is well written.*

AR:  *Thank you for your insightful feedback and suggestions, which we address point-by-point below.*

**2.1.  Minor comment**

RC:  *In Figure 5 it would be helpful to show observations from CERES level 3 data, or from (for example) the Grosvenor et al CDNC dataset, to put the differences between the models in context of the difference between models and observations. Or simply state the typical scale of model-observation discrepancies (likely > 25% in many regions for CDNC at least, I imagine) to give this context, and also provide context to the discussion around line 455.*

AR:  *Thank you for this suggestion. We agree that a comparison to model-observation differences would be helpful here. Unfortunately, these require specific model output. In the case of Grosvenor et al. (2018) these are in-cloud CDNC. As this output is not available with our runs and has already been published for older ECHAM-HAM versions, we instead opted for a comparison to the model versions discussed in Neubauer et al. (2019). With that, we have a more direct comparison, as the new Figure B1 is comparing the exact same diagnostics between the model versions. In fact, in the context of developing a simplified model version that aims to be close to the default version, and not necessarily closer to observations, a comparison to other model versions is more meaningful. For a comparison to observations we refer to Neubauer et al. (2019). For example, their Fig. 1 shows that for cloud-top CDNC the disagreements between model versions are equally large as the deviations from observations. However, they also discuss the caveats in that comparison.*

To address this point we have added Fig. B1 in the manuscript and mention the comparison in the text as follows:

> Still, the deviations are of similar magnitude as differences between successive model versions from Neubauer et al. (2019) (see Fig. B1 in Appendix B and their Figure 1 for a comparison to observations).

**2.2.  Editorial comments**

RC:  *The text in Figure 2 is too small.*

AR:  *We enlarged the Figure.*

RC:  *Would be useful to include a sentence explaining why sections 2.1-2.4 are important to the paper – just a statement that these parameterizations are the ones that relate aerosols to clouds and/or differ between the*

***simulations presented would suffice.***

AR:   *We have added this explanation as suggested.*

> As in the real atmosphere, in ECHAM-HAM the aerosols influence CMP by serving as CCN or INPs in cloud droplet activation and ice crystal nucleation. Since these influences are affected by our simplifications, their implementation in ECHAM-HAM is discussed below. There are two cloud droplet activation parameterizations implemented into ECHAM-HAM, which are both used in this study (see Table A1 for their separate tunings).

**RC:**   *Fig 5 caption needs to define 'mxpT' and 'cold'*

AR:   *Thank you for pointing us to this oversight. We have added the specification.*

> Diagnostics for mixed-phase $(0\,°C > T > -35\,°C)$, liquid $(0\,°C < T)$ and cold $(T < -35\,°C)$ hydrometeor concentrations were computed online.

**RC:**   *L429 believes->beliefs*

AR:   *Thank you for spotting this mistake.*

**RC:**   *The discussion of the relative merits of different possible simplification approaches is more appropriate in the introduction than in the summary; I suggest the authors move lines 431-442 to the introduction.*

AR:   *We welcome this suggestion and have implemented it by modifying and removing passages throughout the text, most importantly adding the following in the introduction:*

We developed a climatology of a) CCN that serves as the connection between the aerosol particles and the CMP and b) aerosol mass and number concentrations. Both climatologies in combination with a pre-existing climatology of aerosol radiative effects can replace the aerosol module HAM. Such a top-down approach, of investigating how the model reacts to changes in aerosols, may be more effective than a tedious bottom-up approach of elaborating all possible processes. Using the climatology allows to isolate remaining processes and their effects and study associated uncertainties. It is important to stress that the climatologies we develop are not meant to be generally applicable. Rather, we propose the process of simplification as a way to gain a new perspective on model behaviour and the simplified model as an explorative tool for further study. This distinguishes our climatologies from the MACv2SP climatology developed by Stevens et al. (2017). Their climatology is analytical, which enhances its flexibility, the clearness of its assumptions and the possibilities for porting it to different models. Our climatology is meant for use in ECHAM with the goal of making its model results resemble those of the default ECHAM-HAM configuration. As such, not the single realisation of our climatology is important, but we rather developed the model code to easily derive and employ new climatologies. This allows the kind of sensitivity studies we use to investigate the SO behaviour (see Section 3.3), which can be helpful in adapting to new model versions and investigating their differences. Further, MACv2SP is restricted to anthropogenic aerosols and prescribes a change to CDNC directly. We take into account all of HAM's (soluble) aerosols. We specifically prescribe either aerosol or potential CCN that enter the cloud droplet activation schemes online and thus keep e.g. the updraft dependence.

This document was generated with a layout template provided by Martin Schrön (github.com/mschroen/review_response_letter).

[revised manuscript text omitted]

---

## Author Response (AR2)

**Author Response to Reviews of**

**Developing a climatological simplification of aerosols to enter the cloud microphysics of a global climate model**

Ulrike Proske, Sylvaine Ferrachat, Ulrike Lohmann
*ACP,* `doi:10.5194/egusphere-2023-2783`
* * *
RC: *Reviewer Comment*,     AR: *Author Response*,     ☐ Manuscript text

We thank the reviewer for the discussion points they raised. We implemented their feedback into a revised version of the manuscript. Please find our answers to the reviewers' points below, followed by a marked-up manuscript verison. Line numbers refer to the previous manuscript version (after the initial round of review).

**1.   Reviewer #1, Comment #2**

RC: *Overall, I'm pleased with the significant improvements made. I have a few points to discuss, including one major concern and some minor suggestions for enhancement.*

RC: *Firstly, the authors may need to further address the application of simplified aerosol treatments. Although I don't doubt effectiveness of simplified aerosol, I believe it's crucial to thoroughly discuss its application in climate change study. As mentioned at Lines of 404-407: "On the other hand, differences between climate states are not affected by the mismatch, and thus the ability of the model to serve in studies of climate change (sensitivity) is preserved in principle. Of course, if one's objective is to study the changes induced by changing aerosol concentrations in detail, a detailed aerosol model will probably suit that purpose better." Here, I find some inconsistency. To my knowledge, the aerosol cooling effect is the most important mechanism that compensates the warming effect of GHGs and it shows uncertainties larger than GHG warming. It implies that both GHG warming and aerosol cooling govern global warming trend. If the Earth system model fails to capture "climate changes induced by changing aerosol concentrations in detail", it raises questions about its suitability for "the ability of the model to serve in studies of climate change (sensitivity) is preserved in principle"?*

AR: *The key to the point you raise are the different modeling visions. Being more specific in terms of formulation, if one's interest is to study aerosol processes in detail and under a representative vision, "a detailed aerosol model will probably suit that purpose better". It is broadly assumed that higher representativeness with added detail will improve projective capabilities as well and thus that a detailed aerosol scheme would serve also the model's predictive purpose better. However, we show that for purely predictive purposes, i.e. looking only at the results of the model, "differences between climate states are not affected by the mismatch". Thus we argue that the simplified model preserves the predictive capabilities well enough for our purpose of studying clouds, at least in terms of changes in global annual means between climate states. We acknowledge that our approach and explicitly formulated model goals differ from the usual approach in the literature, but believe that the teasing apart of the modeling visions and addressing them separately has the potential to make communication clearer in the future. That said, we agree that the lines you quote do not live up to our goal of clear communication and we have thus rephrased them as follows:*

> *On the other hand, differences between climate states are not affected by the mismatch . Regarding the predictive vision, we thus argue that the ability of the model to serve in studies of climate change (sensitivity) is preserved in principle. Of course, if one's objective is to study the changes induced by evolving aerosol concentrations and aerosol processes in detail and under a representative vision, a detailed aerosol model will  suit that purpose better.*

**RC:** *An argument may be like both pre-industrial (PI) and present-day (PD) conditions simulated by simplified aerosol treatment is "similar" to default simulation. However, this may not be enough to state that "the ability of the model to serve in studies of climate change (sensitivity) is preserved in principle". An important measure to examine whether the global warming "global surface temperature anomaly" can be well reproduced [like Figure 23 in Golaz et al. (2022)]. If simplified aerosol treatment is used, how to simulate the global warming trend?*

**AR:** *Evolving aerosol concentrations are represented also in our simplified model version, as the climatology may be constructed from the full model in any aerosol state. The time frequency of the climatology may further be adjusted for higher time resolution, e.g. choosing between a 10 year or 1 year climatology. Thus, if the full model can simulate the global warming trend, the simplified model with input climatologies varying in time should be able to do so as well.*

**RC:** *Additionally, Section 2.6 outlines the setup of PI and PD simulations, but there's a crucial omission regarding the reflection of gas pollutant emission differences between PI and PD conditions. The absence of information on how the model accounts for variations in gas pollutant emissions from pre-industrial to present-day conditions raises concerns about potential underestimation of aerosol forcing. In the default simulation with full aerosol treatment, secondary sulfate aerosols, generated by conversion from SO2 to SO4, are important for the aerosol radiative forcing. If the SO2 emissions in PI and PD conditions are same, the aerosol forcing could be largely underestimated.*

**AR:** *Thank you for pointing us to this omission. We have added the information as follows:*

> Again we keep the applied condition changes simple: for PI simulations we merely use PI  emissions for primary aerosol and gaseous precursors (ACCMIP data (Lamarque et al., 2010), from 1850, 1870, or 1900 as data was available), but everything else (greenhouse gas concentration, SST and sea ice) for the year 2003.

**RC:** *In my view, the simplified aerosol treatment is okay to reflect the climate states, but the suitability for climate change study should be carefully examined. Although I do not entirely oppose simplified aerosol treatment in climate change study, any arguments supporting it must be well-founded and convincing.*

**AR:** *We believe by addressing your points above we have made our argument more clear. Of course, the direction of climate model simplification that we propose would benefit from further study. We agree that diligence must be applied to the assessment of the simplified models' capabilities.*

**1.1. Minor comments**

**RC:** *I suggest adding subtitles (e.g., (a), (b), (c), ...) for each panel to the figures.*

**AR:** *We have implemented this as you suggested.*

**RC:** *Additionally, it would be beneficial to include values for pre-industrial conditions in Table 1, alongside those for present-day conditions.*

**AR:** *We refrain from doing so, because we believe the ARF values in Table 1 together with Fig. 8 and D2 to be sufficient to showcase the results of and differences between the simulations, and do not want to overload the table.*

This document was generated with a layout template provided by Martin Schrön (github.com/mschroen/review_response_letter).

[revised manuscript text omitted]